# Impact of Thermally Inactivated Non-*Saccharomyces* Yeast Derivatives on White Wine

**DOI:** 10.3390/foods13162640

**Published:** 2024-08-22

**Authors:** Valentina Civa, Francesco Maioli, Valentina Canuti, Bianca Maria Pietrini, Matteo Bosaro, Ilaria Mannazzu, Paola Domizio

**Affiliations:** 1Department of Agriculture, Food, Environment and Forestry (DAGRI), University of Florence, 50144 Firenze, Italy; valentina.civa@unifi.it (V.C.); francesco.maioli@unifi.it (F.M.); valentina.canuti@unifi.it (V.C.); bianca.pietrini@stud.unifi.it (B.M.P.); 2Italiana Biotecnologie, Via Vigazzolo 112, 36054 Montebello Vicentino, Italy; matteo.bosaro@italianabiotecnologie.it; 3Department of Agricultural Sciences, University of Sassari, 07100 Sassari, Italy

**Keywords:** wine, mannoprotein, polysaccharides, yeast derivatives, colloidal stability, non-*Saccharomyces*, protein stability

## Abstract

While a recent characterization of non-*Saccharomyces* thermally inactivated yeasts (TIYs) in a wine-like solution highlighted the release of oenologically relevant compounds and different oxygen consumption rates and antioxidant activity, here the impact of TIYs derived from *Saccharomycodes ludwigii* (SL), *Metschnikowia pulcherrima* (MP), *Torulaspora delbrueckii* (TD), and *Saccharomyces cerevisiae* (SC), as the reference strain, was evaluated in white wine. Wine treatment with TIYs resulted in an increase in polysaccharide concentration compared to the untreated wine, with SL-TIY exhibiting the highest release. Additionally, all TIYs, particularly SL-TIY, improved protein stability by reducing heat-induced haze formation. The addition of TIYs also demonstrated an effect on color parameters through phenolic compound adsorption, preventing potential browning phenomena. All TIYs significantly impacted the wine’s volatile profile. Overall, it was shown that an improvement in wine quality and stability may be obtained by using TIYs in the winemaking process.

## 1. Introduction

Yeast derivatives have emerged as valuable tools in winemaking, offering a range of functionalities to enhance wine quality and stability. The growing interest in the use of yeast derivatives as fining agents in winemaking is mainly due to their potential effect on wine quality in terms of color stability [1], oxidative stability [2,3], and the volatile profile [4,5]. Compared to conventional fining agents, yeast derivatives offer several advantages, like their natural origin and the ability to selectively remove undesirable compounds like ochratoxin A [6] and volatile phenols such as ethyl phenol and vinyl phenol [7]. Furthermore, yeast derivatives might represent a biotechnological tool for a faster, more controllable, and more cost-effective alternative to the traditional aging on lees.

Currently, oenological yeast derivatives, authorized by regulations (Resolution OIV-OENO 452-2012; 459-2013; 496-2013; 497-2013; 674-2022) and commercially available, are obtained from *S. cerevisiae*. Several scientific studies have highlighted the impact of *S. cerevisiae* yeast derivatives on wine quality [4,8,9,10]. Less is known about derivatives from non-*Saccharomyces* yeasts, despite the renewed interest in their role as fermentation starters in winemaking [11,12,13]. Yeasts belonging to different genera, species, and strains present metabolic differences [14,15], and differ in cell wall composition [16]. Both whole cells and cellular fractions exert various effects on wine’s chemical-physical composition, mainly due to the intrinsic characteristics of mannoproteins [17,18,19,20]. The mannose–glucose ratio in mannoproteins can impact wine protein stability [20], while the presence of mannosylphosphate groups seems to improve the polyphenol adsorption by yeast cell walls, modulating wine color and astringency [21].

In a recent study, Civa et al. [22] reported that non-*Saccharomyces* TIYs in a wine-like solution release soluble compounds like polysaccharides, lipids, thiols, and reduced glutathione (GSH) and exhibit varying oxygen consumption rates and antioxidant activity. 

The present study aims to gather further information on the biotechnological potential of non-*Saccharomyces* TIYs by investigating their effects on the chemical-physical characteristics of white wine. For that, the effect of non-*Saccharomyces* TIYs on protein stability, color indexes, volatile profile, and polysaccharide concentration was evaluated 15 days after TIYs addition to a Trebbiano Toscano white wine.

## 2. Materials and Methods

### 2.1. Yeast Strains and TIYs Preparation

The yeast strains utilized for the production of TIYs are listed in Table 1. The commercial strain of *S. cerevisiae*, Lalvin EC1118 (Lallemand—Montreal, QC, Canada), was used as a reference strain. Non-*Saccharomyces* yeasts, as described in [12,14], are deposited in the microbial culture collection of DAGRI. TIYs production was carried out as described in Civa et al. [22]. Briefly, yeast pre-cultures were initially grown in 75 mL of a growth medium (2.5% yeast extract, 2% peptone, 5% glucose, and 5% fructose) within 100 mL flasks. Incubation occurred at 27 °C for 24 h in an orbital shaker at 150 rpm. Subsequently, 1% of each pre-culture was transferred to flasks containing 750 mL of the same medium and incubated for 72 h under identical conditions. Then, cultures underwent centrifugation at 4 °C for 8 min at 8000 rpm. The resulting cell pellets were washed thoroughly and resuspended in sterile distilled water to achieve a 1:5 biomass/distilled water ratio (*w*/*v*). Thermal inactivation was performed at 121 °C for 1 h, and the inactivated yeast biomass was freeze-dried, yielding TIY powder.

### 2.2. Wine Treatment

Wine of the variety *Vitis vinifera* cv. Trebbiano Toscano (vintage 2022) provided by Cantina Cooperativa Colli Fiorentini (Montespertoli, Firenze, Tuscany, Italy) was used. The wine was filtered through a Jumbo-Star Sartopure equipped with a 0.45 µm polypropylene cartridge (Sartorius Stedim Biotech GmbH, Göttingen, Germany), and no stabilization treatments were carried out. The oenological standard parameters and protein stability value, obtained by the heat test (ΔNTU) (see 1.4), are reported in Table 2. Wine samples (200 mL) were added with TIYs (40 g/hL). Analyses were carried out after 15 days of treatment at 20 ± 2 °C. Wine samples w/o TIY were used as controls (CT). All the trials were set up in triplicate.

### 2.3. Total Polysaccharides

Total polysaccharides were measured using high-performance liquid chromatography (HPLC), following the protocol outlined in Millarini et al. [23]. In brief, 20 µL of each sample were injected into the HPLC system (Varian Inc., Palo Alto, CA, USA), featuring a 410 series autosampler, a 210 series pump, and a 356-LC refractive index (RI) detector. Isocratic separation was conducted on a TSKgel OLIGO-PW (808031) column (30 cm × 7.8 mm i.d.) coupled with a TSKgel OLIGO (808034) guard column (4 cm × 6 mm i.d.) (Supelco, Bellefonte, PA, USA). The mobile phase consisted of 0.2 M sodium chloride (Sigma-Aldrich, Milan, Italy), flowing at a rate of 0.8 mL/min. Peak quantification was achieved by referencing an external calibration curve established using mannan (Sigma-Aldrich, Milan, Italy) across concentrations ranging from 50 to 1000 mg L^−1^. Peak integration was performed by Galaxie Chromatography Data System software (version 1.9.302.530) (Varian Inc., Palo Alto, CA, USA). All analyses were conducted in triplicate.

### 2.4. Heat Test

The protein stability of the wine samples was assessed to understand the impact of the TIYs. The induced haze value following the heat test was determined according to Pocock and Waters [24]. Wine aliquots were filtered using 0.45 μm acetate cellulose membranes and then subjected to heating at 80 °C for 2 h. Subsequently, the aliquots were cooled at 4 °C for 16 h and left at room temperature for 2 h before measuring their turbidity using a nephelometer (HI88703 turbidimeter, Hanna Instrument Inc., Woonsocket, RI, USA).

### 2.5. Colour Indexes and CIEL*a*b* Trichromatic Coordinates

CIE (Commission Internationale de l’Eclairage) L* (lightness, 0 black and 100 white), a* and b* (red/green; yellow/blue) color coordinates, and A420, A520, and A620 were measured by SmartAnalysis instrument (DNAPhone, Parma, Italy). Color intensity (CI), A420/A520 tone (T), chroma (C*), and hue* (tone) were also calculated. Color differences between wines were determined using the ∆E value calculated as the Euclidean distance between two points (1 and 2) in three-dimensional (L*, a*, b*) space, according to the following Equation (1):(1)$$∆E=\sqrt{{\left(\Delta L*\right)}^{2}+{\left(\Delta a*\right)}^{2}+{\left(\Delta b*\right)}^{2}}$$

When ∆E > 3, differences between wines are perceivable by human sight [25]. All the analyses were performed in triplicate.

### 2.6. Phenolic Compounds

Total phenolic compounds were quantified by HPLC [26]. The analysis was carried out on a Perkin-Elmer Series 200 LC equipped with an autosampler and a diode-array detector (DAD Series 200) (Perkin-Elmer—Waltham, MA, USA). Chromatograms were acquired at 280 nm, recorded, and processed using Total Chrome Navigator software (V 6.2.1) (Perkin-Elmer—Waltham, MA, USA). A polystyrene divinylbenzene column (250 mm × 4.6 mm PLRP-S 100A 5 μm, Polymer Laboratories Inc., Amherst MA, USA) was used with a guard cartridge (10 × 4.6 mm) packed with the same material (both from Lab Service Analytica Srl, Bologna, Italy). The column was held at 28 °C. Wines were filtered at 0.22 μm with an acetate cellulose syringe filter before injection. One mL of sample was collected in 2 mL HPLC vials with an addiction of 10 μL of formic acid. The volume injected was 20 μL, with the binary pump flow set at 1 mL/min using the following eluents: (A) a water solution of 1.5% (*w*/*w*) of ortho-phosphoric acid; (B) 20% of (A) in acetonitrile. Eluent gradients were set as follows: for the first 55 min, from 92% to 73% of eluent A, maintaining the isocratic conditions of 73% from minute 55 to 59, reducing from 73% to 30% between 59 and 64 min, maintaining at 30% from minute 64 to 69, and increasing to 92% from 70 to 76 min. Total phenols were calculated by the sum of all of the peak areas detected and expressed as mg/L of gallic acid equivalent. Reagents and standards (gallic acid, (+)-catechin, (−)-epicatechin, *p*-cumaric acid, procyanidin B1, and caffeic acid) were purchased by Sigma-Aldrich (Milan, Italy). Compounds were identified based on the retention time and UV–visible spectra of the standard compounds injected (Appendix A). 

### 2.7. Volatile Compounds

To extract volatile compounds from the samples, a solid-phase extraction (SPE) process was conducted using polystyrene-divinylbenzene copolymer (PS/DVB) SPE cartridges (Macherey-Nagel CHROMABOND Easy 3 mL/200 mg), which were initially conditioned with 3 mL of a water–ethanol solution (6% *v*/*v*). Subsequently, 25 mL of wine underwent filtration at 0.2 µm and was then diluted in a 1:1 ratio with water. Following this, 40 µL of 2-octanol internal standard (6.3 mM in ethanol) was added (Sigma-Aldrich, Milan, Italy). The sample was passed through the SPE cartridge at a rate of approximately 2 mL/min, after which the sorbent was dried by passing air through it. The analytes were eluted using two aliquots of dichloromethane (2 × 1.0 mL) (Sigma-Aldrich, Milan, Italy). The sealed samples were then analyzed by GC-MS, utilizing a GC Perkin-Elmer Clarus 580 instrument coupled with a Perkin-Elmer SQ8S MS detector (Perkin-Elmer—Waltham, MA, USA). The capillary column employed was a Perkin-Elmer WAX-ETR (30 m × 0.32 mm ID × 0.25 µm), with helium used as the carrier gas at a flow rate of 1.5 mL/min. The gas chromatograph temperature ramp began at 40 °C for 1 min, followed by an increase of 5 °C/min up to 240 °C, which was maintained for 5 min. A 1 µL sample injection was performed via an autosampler. The injector (SPLIT) was set at a temperature of 250 °C with a flow rate of 20 mL/min.

### 2.8. Data Analysis

The collected data underwent an analysis of variance (ANOVA), followed by Tukey’s honest significant difference test at a significance level of 0.05 to assess differences between the data sets. The means, along with their standard deviations (mean ± SD), are provided. Differences and similarities among the samples were studied by means of principal component analysis (PCA) and hierarchical cluster analysis. Statistical analysis was performed using the XLSTAT software package (version 2023.3.1, Addinsoft, Paris, France). 

## 3. Results and Discussion

### 3.1. Total Polysaccharides content following TIYs treatment

Wines treated with the different TIYs and the control untreated wine (CT) were characterized for total polysaccharide content (Figure 1). Treated wines showed higher concentrations of polysaccharides compared to the CT (227.96 mg/L ± 3.5). In particular, the highest concentration was detected in SL-TIY wine (304.15 mg/L ± 10.6), followed by TD-TIY (296.62 mg/L ± 3.7), MP-TIY (271.21 mg/L ± 5.8), and SC-TIY (255.83 mg/L ± 3.2). Enrichment in total polysaccharides after *S. cerevisiae* yeast derivative addition has already been reported [4,27]. Accordingly, SC-TIY wine contained 12% more polysaccharides than CT. Notably, polysaccharides in non-*Saccharomyces* TIY-treated wines were 19–33% higher than those of CT. This result agrees with that already observed in a wine-like solution [22]. The polysaccharides released by the TIYs after heat treatment are likely those extracted from the cell wall and not covalently bound [28]. Therefore, β-glucanase treatment [29] or high-pressure techniques [30] could be useful to accelerate the release of covalently bound polysaccharides by TIYs.

### 3.2. Protein Stability of TIYs treated wines

Protein stability of the experimental Trebbiano Toscano wines was assessed 15 days after the addition of the TIYs to evaluate their potential effect on colloidal stability in comparison with the control untreated wine (CT). The impact of the TIYs on heat-induced protein haze formation was evaluated by nephelometry and reported as the difference in nephelometric turbidity units (ΔNTU) between heated and unheated samples (Figure 2).

The CT showed a ΔNTU of 10.4 ± 0.5. With the exception of MP-TIY, all the TIY-treated wines presented a significant decrease in ΔNTU and therefore a decrease in haziness induced by heating. Despite previous studies showing that polysaccharides, in particular mannoproteins, might positively impact wine protein stability [23,31,32], no clear correlation was found here between the amount of polysaccharides released and protein stability. In particular, SC-TIY, which showed the lowest amount of polysaccharides (255.83 mg/L), resulted in a 26% decrease in protein haze. Instead, MP-TIY (271.21 mg/L) and TD-TIY (296.62 mg/L) showed a wine protein stability improvement with a reduction of protein haze of 15% and 20%, respectively. Instead, SL-TIY (304.15 mg/L) resulted in the highest reduction in protein haze induced by heating and improved Trebbiano Toscano protein stability by 32%. 

The polysaccharides released by each TIY differed in concentration, molecular weight profiles, and mannose/glucose ratio, likely due to variations in yeast biodiversity and growth stages during inactivation [22]. According to Ribeiro and colleagues [20], mannoproteins characterized by a high mannose/glucose ratio appear to improve protein stability in wine. However, other mechanisms, such as interaction/competition with other wine compounds, seem to be involved in the improvement in protein stability [23,33]. The TIYs here utilized affected Trebbiano Toscano wine protein stability to different extents. However, the degree of stabilization attained with these derivatives is not sufficient to significantly reduce the bentonite doses needed to obtain ΔNTU values < 2 [23].

### 3.3. Impact of TIYs on Colour Indexes, CIEL*a*b* Coordinates and Phenol Content

Color indexes (CI) and CIEL*a*b coordinates of TIY-treated wines showed significant differences with respect to the control untreated wine (CT) (Table 3). Concerning the spectrophotometric indexes, MP-TIY wine showed significantly lower values (*p* < 0.5) of A420, A520, and A620 (and consequently in CI), followed by TD-TIY wine compared to CT. Although these parameters gather interesting information about wine color, the measurement of CIEL*a*b coordinates allows for its real evaluation. A general decrease in the color of the white wines treated with non-*Saccharomyces* TIYs was observed. These were characterized by higher L* values (luminosity) and lower values of both a* (green-red) and b* (blue-yellow) coordinates with respect to CT. Of all the TIY-treated wines, MP- and TD-TIYs wines showed significantly lower values of Chroma* (color vividness). However, the color differences in the TIY-treated wines with respect to the CT could not be perceivable to the human eye due to ΔE values < 3. It is worth mentioning that color evaluation, including CIEL*a*b coordinates, is usually considered an indicator of wine color stability, and change is likely due to oxidation processes [34]. 

In general, total phenol content was in line with spectrophotometric indexes (Table 3). Indeed, with the exception of SC-TIY wine, all non-*Saccharomyces* TIY treated wines showed a significantly lower total phenol content than the CT. In particular, trans-caftaric acid, p-coumaric acid, caffeic acid, and procyanidin B1 were the phenolic compounds most affected by the TIY treatment (Appendix A). In agreement, phenolic compound adsorption by yeasts, yeast cell walls, and inactivated yeast has already been reported [18,19]. Hence, TIYs might contribute to preventing white wine browning caused by the oxidation of phenolic compounds, particularly flavan-3-ol derivatives and hydroxycinnamic acids [34,35]. It is worth highlighting that the release of other compounds with antioxidant properties, such as reducing compounds (i.e., GSH) and lipids, could contribute to maintaining wine’s oxidative stability and preserving its sensory characteristics after exposure to oxygen [3,22]. 

### 3.4. Effect of TIYs on volatile compounds

Volatile compounds of wine samples treated with the TIYs, determined by SPE-GC/MS, showed a general reduction across samples (Table 4). Significantly lower values of several compound families were detected in TIY-treated wines with respect to the CT. In particular, this effect was detected for alcohols (2-phenyl ethanol; 1-hexanol; benzyl alcohol; trans—(3)-hexenol; 3-ethoxy propanol; 3-methyl thiopropanol); terpenoids (*α*-terpineol; geraniol); fatty acids (2-tethyl propanoic acid; butanoic acid; 3-methyl butanoic acid); fatty acid ethyl esters (ethyl 3-methyl butanoate; ethyl decanoate; ethyl 2-phenyl acetate); carboxylic acid ethyl esters (mono ethyl succinate; diethyl succinate); and total aldehydes in terms of benzaldehyde. Moreover, SL-TIY and MP-TIY wines were characterized by a significantly higher content of n-butyl acetate in comparison to the TD-, SC-TIY, and CT wines (Table 3). This could be due to lower adsorption by these TIYs.

Volatile compound families were processed with PCA analysis to highlight differences among wines (Figure 3). The total explained variance was 94.40% (PC1 63.17%, PC2 31.23%), and the wines were separated for their volatiles accordingly. CT wine is in the 1st quadrant of the PCA characterized by the highest amount of volatile compound families, whereas all the other wines were distributed on the opposite side of the first dimension (negatively correlated). Furthermore, wines were also separated by the PC2 for fatty acid content, where TD- and SL-TIYs wines were negatively correlated with respect to the CT wine.

Previous research indicated that the influence of yeast derivatives on the volatility of wine compounds can vary widely, depending on the specific volatile compound and the yeast derivative used [36,37,38]. Prolonged contact with dry yeast derivatives may amplify this effect, as yeast cellular components and compounds continuously interact over time, potentially altering the volatile profile of wine [1]. Considering the complexity of the wine matrix, which arises from a multitude of compounds present at different concentrations (i.e., a high concentration of 3-methylbutanol), competitive interactions could occur at the binding sites on the cell wall, thus influencing the retention of specific volatiles. The influence of yeast derivatives, like TIYs, on volatile compounds is mainly attributed to their possible adsorption by the yeast cell wall through hydrophobic interactions. This adsorption might also be explained by the presence of lipids in the cell wall [39]. Indeed, other authors have shown different behavior in the interaction of volatile compounds with mannoproteins. Chalier et al. [38] highlighted that compounds such as ethyl-hexanoate have greater affinity for the glycosidic rather than proteic parts of mannoproteins. Langourieux and Crouzet [40] investigated the binding capabilities of various polysaccharides and reported a salting-out effect with limonene and ethyl hexanoate water solutions upon the addition of high-molecular-weight dextran. As reported by Civa et al. [22], SL-TIY can release polysaccharides composed mainly of mannose and characterized by a high molecular weight in comparison to the ones released by SC-TIY. Therefore, the biodiversity of yeast derivatives and the differences in their intrinsic composition might be used to selectively retain volatile compounds, thus affecting the overall wine volatile profile. However, it is worth considering that other variables, such as the wine pH and temperature, were reported to have an impact on yeast colloidal interactions with several volatile compounds [36].

## 4. Conclusions

As far as is known, this is the first time the impact of non-*Saccharomyces* thermally inactivated yeasts has been evaluated in a real white wine. Observed positive effects included wine protein stabilization and browning prevention. Notably, the effect of TIY polysaccharides on protein stabilization was independent of the amount of polysaccharides released. Therefore, further studies are needed to explore the mechanisms underlying the interactions between these polysaccharides and the overall wine colloidal matrix. Instead, the adsorption of phenolic compounds by TIYs might be responsible for preventing white wine browning, thereby satisfying visual appeal characteristics. The impact of TIYs on the volatile aroma compounds within wine samples varies depending on the specific compound and the yeast derivative used. However, it is worth mentioning that the addition of TIYs showed a significant effect on the most important families of volatile compounds, although sensory analyses are required to confirm this result. Overall, the non-*Saccharomyces* TIYs, as well as SC-TIY, appear as useful winemaking bio-adjuvants, although further research is needed to investigate the utilization of TIYs in different stages of the winemaking process.

## Figures and Tables

**Figure 1 foods-13-02640-f001:**
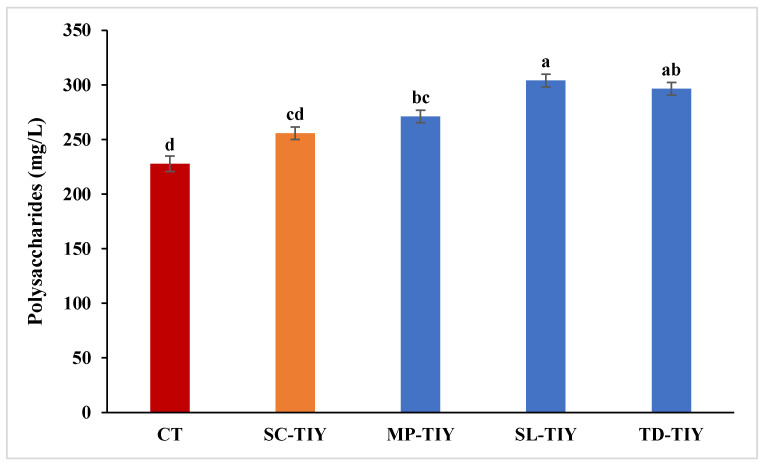
Total polysaccharide concentration (mg/L) in Trebbiano Toscano wine samples 15 days after the addition of TIYs. CT: control untreated wine; SC-TIY: *S. cerevisiae* TIY; MP-TIY: *M. pulcherrima* TIY; SL-TIY: *S. ludwigii* TIY; TD-TIY: *T. delbrueckii* TIY. Data are average ± standard deviation of three independent replicates. Different letters indicate values significantly different. LSD, least significant difference test; 95% significance level.

**Figure 2 foods-13-02640-f002:**
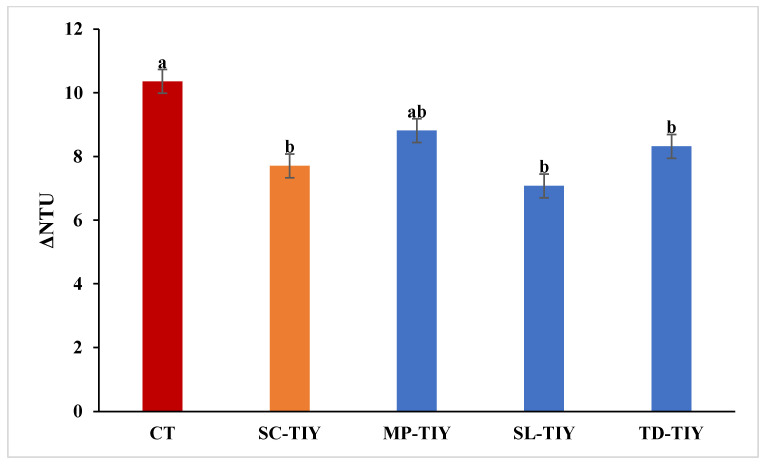
Haziness of Trebbiano Toscano wine supplemented with TIYs as obtained by nephelometry after heating treatment. CT: control untreated wine, SC-TIY: *S. cerevisiae* TIY; MP-TIY: *M. pulcherrima* TIY, SL-TIY: *S. ludwigii* TIY, TD-TIY: *T. delbrueckii* TIY. Data are average ± standard deviation of three independent replicates. Different letters indicate values significantly different. LSD, least significant difference test; 95%, significance level.

**Figure 3 foods-13-02640-f003:**
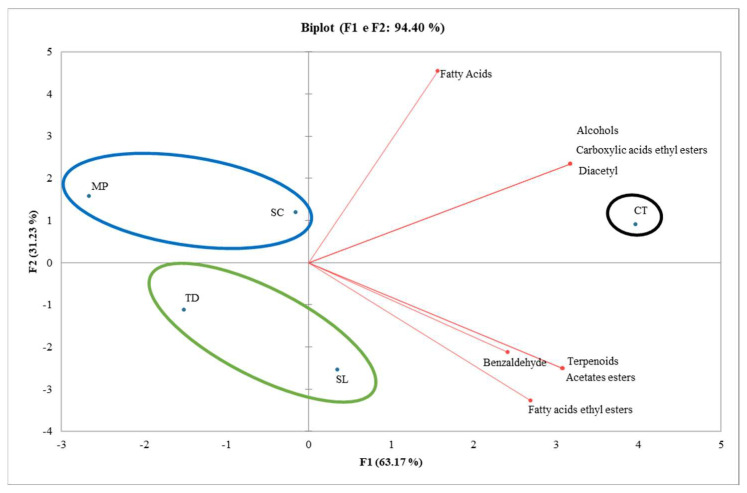
Principal component analysis (PCA): scores and loadings bi-plot of the volatile family compounds (alcohols; terpenoids; diacetyl and benzaldehyde; fatty acids; fatty acids ethyl esters; carboxylic acid ethyl esters; acetate esters) of wines added with TIYs.

**Table 1 foods-13-02640-t001:** Yeast strains utilized for TIYs production.

Strain	Species	Origin	TIY Code
EC1118	*Saccharomyces cerevisiae*	Lallemand ^a^	SC-TIY
46	*Metschnikowia pulcherrima*	DAGRI ^b^	MP-TIY
64	*Saccharomycodes ludwigii*	DAGRI ^b^	SL-TIY
92	*Torulaspora delbrueckii*	DAGRI ^b^	TD-TIY

^a^ Lallemand Inc. (Montreal, QC, Canada). ^b^ Department of Agriculture, Food, Environment and Forestry (DAGRI), University of Florence, Italy.

**Table 2 foods-13-02640-t002:** Oenological standard parameters (alcohol, titratable acidity, volatile acidity, pH) and haziness value (ΔNTU) of Trebbiano Toscano white wine (average ± standard deviation).

Wine	Alcohol % (*v*/*v*)	Titratable Acidity (Tartaric Acid g/L)	Volatile Acidity (Acetic Acid g/L)	pH	ΔNTU
Trebbiano Toscano	12.18 ± 0.02	5.76 ± 0.02	0.13 ± 0.02	3.14 ± 0.01	10.40 ± 0.54

**Table 3 foods-13-02640-t003:** Total phenols (TP), color indexes and CIEL*a*b* coordinates of Trebbiano Toscano white wines added with TIYs.

	CT			SC-TIY			MP-TIY			SL-TIY			TD-TIY		
A420	0.099 ^a^	±	0.001	0.084 ^b^	±	0.005	0.075 ^d^	±	0.001	0.084 ^b^	±	0.002	0.080 ^c^	±	0.002
A520	0.038 ^a^	±	0.001	0.029 ^b^	±	0.004	0.019 ^c^	±	0.001	0.028 ^b^	±	0.001	0.027 ^b^	±	0.001
A620	0.011 ^a^	±	0.001	0.012 ^a^	±	0.004	0.003 ^b^	±	0.001	0.011 ^a^	±	0.001	0.011 ^a^	±	0.001
CI	0.149 ^a^	±	0.003	0.125 ^b^	±	0.013	0.098 ^c^	±	0.001	0.122 ^b^	±	0.004	0.118 ^b^	±	0.004
Tone	2.620 ^c^	±	0.037	2.991 ^b^	±	0.252	3.965 ^a^	±	0.037	2.986 ^b^	±	0.07	2.982 ^b^	±	0.074
Chroma*	6.633 ^a^	±	0.018	5.564 ^b^	±	0.098	5.303 ^c^	±	0.048	5.584 ^b^	±	0.054	5.354 ^c^	±	0.032
Hue (°)	87.536 ^d^	±	0.479	93.073 ^b^	±	0.414	94.097 ^a^	±	0.251	92.40 ^c^	±	0.285	92.947 ^b^	±	0.251
L*	97.411 ^c^	±	0.078	97.911 ^b^	±	0.362	98.44 ^d^	±	0.053	97.967 ^b^	±	0.1	98.011 ^b^	±	0.093
a*	0.287 ^a^	±	0.04	−0.299 ^c^	±	0.036	−0.384 ^d^	±	0.013	−0.231 ^b^	±	0.028	−0.272 ^c^	±	0.015
b*	6.626 ^a^	±	0.018	5.554 ^b^	±	0.098	5.288 ^c^	±	0.047	5.579 ^b^	±	0.051	5.346 ^c^	±	0.034
ΔE				1.32			2.00			1.29			1.52		
TP ^1^	66.680 ^a^	±	0.574	66.080 ^ab^	±	0.529	61.025 ^bc^	±	2.286	60.196 ^bc^	±	1.806	55.898 ^d^	±	3.339

Data are average ± standard deviation of three independent replicates. Different letters in the same row indicate values significantly different. LSD, least significant difference test; 95%, significance level. ^1^ mg/L of gallic acid equivalent.

**Table 4 foods-13-02640-t004:** Volatile compounds content of TIY-treated wines.

Compound	CT			SC-TIY			MP-TIY			SL-TIY			TD-TIY		
Alcohols (mg/L)															
2-Methylpropanol	13.60 ^a^	±	0.20	13.58 ^a^	±	1.00	13.91 ^a^	±	0.55	14.66 ^a^	±	0.26	13.36 ^a^	±	0.13
n-Butanol	0.07 ^a^	±	0.00	0.06 ^a^	±	0.00	0.06 ^a^	±	0.00	0.07 ^a^	±	0.01	0.07 ^a^	±	0.01
2-Methylbutanol	15.93 ^a^	±	0.26	15.00 ^a^	±	0.73	15.39 ^a^	±	1.07	14.17 ^a^	±	0.84	14.59 ^a^	±	0.51
3-Methylbutanol	99.51 ^a^	±	15.53	92.03 ^a^	±	1.08	82.54 ^a^	±	0.44	84.22 ^a^	±	2.65	80.92 ^a^	±	2.06
2-Phenylethanol	49.18 ^a^	±	5.61	41.08 ^ab^	±	1.9	41.52 ^ab^	±	0.4	39.76 ^b^	±	2.42	41.79 ^ab^	±	1.48
1-Hexanol	1.03 ^a^	±	0.05	0.89 ^b^	±	0.01	0.85 ^b^	±	0.00	0.88 ^b^	±	0.05	0.97 ^ab^	±	0.01
Benzyl alcohol	0.41 ^a^	±	0.01	0.34 ^b^	±	0.00	0.33 ^b^	±	0.01	0.33 ^b^	±	0.02	0.35 ^b^	±	0.02
Trans-(E)-3-Hexenol	0.05 ^a^	±	0.00	0.04 ^c^	±	0.00	0.04 ^c^	±	0.00	0.04 ^c^	±	0.00	0.05 ^b^	±	0.00
Cis-(Z)-3-Hexenol	0.24 ^a^	±	0.02	0.21 ^a^	±	0.00	0.20 ^a^	±	0.00	0.21 ^a^	±	0.01	0.23 ^a^	±	0.00
3-Ethoxypropanol	0.09 ^a^	±	0.01	0.07 ^ab^	±	0.01	0.07 ^ab^	±	0.00	0.06 ^b^	±	0,00	0.07 ^ab^	±	0.00
3-Methylthiopropanol	2.30 ^a^	±	0.01	2.13 ^b^	±	0.00	2.30 ^a^	±	0.02	2.09 ^bc^	±	0.02	2.03 ^c^	±	0.02
Total alcohols	182.41			165.41			157.20			156.50			154.42		
Terpenoids (μg/L)															
Linalool	16.04 ^a^	±	0.40	14.24 ^a^	±	2.04	13.16 ^a^	±	0.06	13.16 ^a^	±	0.85	14.96 ^a^	±	0.23
α-Terpineol	6.44 ^a^	±	0.51	4.56 ^b^	±	0.23	4.48 ^b^	±	0.34	4.20 ^b^	±	0.28	5.08 ^b^	±	0.06
β-Citronellol	20.33 ^a^	±	4.68	18.29 ^a^	±	5.84	17.99 ^a^	±	0.75	18.14 ^a^	±	0.57	15.55 ^a^	±	4.20
Nerol	32.36	±	1.98	29.80 ^a^	±	2.43	29.20 ^a^	±	1.92	27.92 ^a^	±	1.92	27.24 ^a^	±	2.55
Geraniol	12.06 ^a^	±	0.49	9.00 ^b^	±	0.03	9.35 ^b^	±	0.04	10.05 ^ab^	±	0.31	10.71 ^ab^	±	1.17
β-Damascenone	11.32 ^a^	±	2.43	7.68 ^a^	±	0.68	8.68 ^a^	±	0.40	7.92 ^a^	±	0.45	8.20 ^a^	±	0.06
β-Ionone	0.80 ^a^	±	0.00	0.70 ^a^	±	0.14	0.60 ^a^	±	0.00	0.80 ^a^	±	0.00	0.90 ^a^	±	0.14
Total terpenoids	99.35			84.27			83.46			82.19			82.64		
Carbonyl compounds (mg/L)															
Benzaldehyde	93.46 ^a^	±	6.31	89.76 ^a^	±	4.07	75.32 ^b^	±	0.40	101.94 ^a^	±	1.50	75.44 ^b^	±	2.04
Diacetyl	0.18 ^ab^	±	0.02	0.14 ^b^	±	0.01	0.19 ^a^	±	0.00	0.18 ^ab^	±	0.01	0.22 ^a^	±	0.01
Total carbonyl compounds	93.64			89.90			75.51			102.12			75.66		
Fatty acids (mg/L)															
2-Methylpropanoic acid	1.98 ^a^	±	0.08	1.82 ^abc^	±	0.02	1.87 ^ab^	±	0.00	1.68 ^c^	±	0.03	1.71 ^bc^	±	0.01
Butanoic acid	1.26 ^a^	±	0.00	1.01 ^b^	±	0.02	1.01 ^b^	±	0.08	0.93 ^b^	±	0.03	0.96 ^b^	±	0.02
3-Methylbutanoic acid	0.37 ^a^	±	0.02	0.31 ^ab^	±	0.03	0.31 ^ab^	±	0.00	0.28 ^b^	±	0.02	0.30 ^b^	±	0.00
Hexanoic acid	9.38 ^a^	±	1.11	10.16 ^a^	±	0.49	9.37 ^a^	±	0.84	9.39 ^a^	±	0.25	8.39 ^a^	±	0.88
Octanoic acid	14.36 ^a^	±	1.85	13.78 ^a^	±	0.78	13.67 ^a^	±	0.46	12.55 ^a^	±	0.44	12.72 ^a^	±	0.02
Decanoic acid	1.05 ^a^	±	0.11	0.76 ^a^	±	0.11	0.90 ^a^	±	0.11	0.92 ^a^	±	0.10	0.80 ^a^	±	0.08
Total fatty acids	28.41			27.84			27.12			24.71			24.99		
Fatty acids ethyl esters (μg/L)															
Ethyl isobutyrate	42.04	±	2.39	38.13	±	2.64	37.20	±	2.07	41.91	±	1.70	40.40	±	0.57
Ethyl butanoate	588.63	±	64.71	537.24	±	7.07	491.88	±	20.31	543.56	±	7.64	530.92	±	13.18
Ethyl 3-methylbutanoate	17.20 ^a^	±	1.36	14.60 ^ab^	±	0.17	13.32 ^b^	±	1.19	14.64 ^ab^	±	0.23	16.48 ^ab^	±	0.57
Ethyl hexanoate	905.08	±	129.49	740.52	±	40.56	789.64	±	31.17	825.32	±	14.99	856.24	±	35.07
Ethyl octanoate	1058.88 ^ab^	±	53.51	1026.12 ^ab^	±	102.11	880.80 ^b^	±	48.08	1233.04 ^a^	±	40.28	1064.68 ^a^	±	18.27
Ethyl decanoate	214.22 ^a^	±	13.83	139.11 ^b^	±	18.60	106.40 ^b^	±	22.88	163.20 ^ab^	±	0.38	152.53 ^b^	±	3.14
Ethyl 2-phenylacetate	9.96 ^a^	±	0.62	8.12 ^b^	±	0.28	8.48 ^ab^	±	0.57	7.64 ^b^	±	0.17	8.36 ^ab^	±	0.06
Total fatty acids ethyl esters	2836.35			2503.84			2327.72			2829.31			2669.61		
Carboxylic acid ethyl esters (mg/L)															
Ethyl lactate	2,06 ^a^	±	0.24	1.91 ^a^	±	0.13	2.03 ^a^	±	0.11	1.98 ^a^	±	0.07	2.01 ^a^	±	0.01
Monoethyl succinate	49.90 ^a^	±	1.38	34.25 ^c^	±	2.37	41.12 ^b^	±	1.49	39.76 ^bc^	±	0.08	41.25 ^b^	±	0.39
Diethyl succinate	2.58 ^a^	±	0.13	2.28 ^b^	±	0.07	2.28 ^b^	±	0.01	2.13 ^b^	±	0.03	2.20 ^b^	±	0.02
Total carboxylic acids esters	54.54			38.45			45.44			43.86			45.46		
Acetate esters (μg/L)															
Isobutyl acetate	50.81 ^a^	±	1.13	48.2 ^a^	±	2.98	50.25 ^a^	±	0.36	52.18 ^a^	±	1.73	51.42 ^a^	±	0.77
n-butyl acetate	9.07 ^b^	±	0.91	9.01 ^b^	±	0.96	27.55 ^a^	±	0.40	26.59 ^a^	±	0.68	12.56 ^b^	±	0.74
Isoamyl acetate	2305.81 ^a^	±	78.30	2263.41 ^a^	±	76.22	2241.14 ^a^	±	7.68	2350.00 ^a^	±	28.11	2216.17 ^a^	±	83.90
n-hexyl acetate	33.64 ^a^	±	2.60	28.08 ^a^	±	3.39	29.12 ^a^	±	0.51	33.08 ^a^	±	1.19	29.76 ^a^	±	1.92
2-phenylethyl acetate	990.84 ^a^	±	106.86	1023.73 ^a^	±	23.53	1010.64 ^a^	±	38.52	999.56 ^a^	±	7.52	962.96 ^a^	±	4.53
Total acetates	3390.16			3372.43			3358.69			3461.53			3272.87		

Data are average ± standard deviation of three independent replicates. Different letters in the same row indicate values significantly different. LSD, least significant difference test; 95%, significance level.

## Data Availability

The original contributions presented in the study are included in the article/Appendix A, further inquiries can be directed to the corresponding authors.

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
