# Peer review of "Impact of Thermally Inactivated Non-*Saccharomyces* Yeast Derivatives on White Wine"

_foods, 2024, doi:10.3390/foods13162640_

Round 1

Reviewer 1 Report

Comments and Suggestions for Authors

In my opinion this paper is very interesting because it is the first time that the impact of non Saccharomyces thermally inacetivated yeast in a real withe wine has been evaluated. This effect has been studied in terms of total polysaccharides content, colour indexes and volatile composition.

In addition, statistical analyses were carried out and PCA figure is included in this paper.

However, I want to ask authors about the wine chosen for this study. Why do you choose it? And what is the reason for which do you study only one sample wine? Authors have to justify this point, because only one sample wine has been analyzed.

Conclusions are very well written and authors comment that sensory analyses are required in order to confirm the conclusions reached in this paper.

Author Response

In my opinion this paper is very interesting because it is the first time that the impact of non Saccharomyces thermally inactivated yeast in a real withe wine has been evaluated. This effect has been studied in terms of total polysaccharides content, colour indexes and volatile composition.

In addition, statistical analyses were carried out and PCA figure is included in this paper. However, I want to ask authors about the wine chosen for this study. Why do you choose it? And what is the reason for which do you study only one sample wine? Authors have to justify this point, because only one sample wine has been analyzed.

Trebbiano Toscano is one of the main and most widespread white grape varieties in Tuscany, and it is notably rich in polyphenols, which can cause oxidative color instability. For this reason, we believe it is ideal for our study. This exploratory work has highlighted key aspects to focus on, and we believe it will soon be possible to evaluate the impact of inactivated yeasts on other wine types and a larger number of samples.

Conclusions are very well written and authors comment that sensory analyses are required in order to confirm the conclusions reached in this paper.

Reviewer 2 Report

Comments and Suggestions for Authors

Dear authors,

you manuscript deals with an interesting subject, and you did a big research. Therefore, it should be presented in its best quality. I have few recommendation:

- In the abstract, line 13/14 - after Saccharomyces cerevisiae (SC), I recommend adding: "as reference strain". When reading only abstract, a reader could ask how is this non-Saccharomyces yeast, while the title indicates that the study is only about them.

- I recommend adding some information about thermally inactivated yeasts; why? how?, so readers could better understand the purpose of the study.

- Phenolic compounds determination by HPLC: Is there a reason why no individual phenolic compounds were identified with HPLC, only the total sum?

- Volatile compounds determination by GC-MS: Could you please explain what was the peak identification method (retention time, retention index, or mass spectra base)?

- How did you calculate the concentrations of volatile compounds? Was the response ratio for each compound calculated in comparison to the internal standard, or the calculation was tentatively using only peak area and concentration of internal standard? This could also be added in the text if possible.

- Table 4. Volatile compounds: Why only total aldehydes are presented, and no individual compounds? How was this obtained?

- Aldehydes are also classified as carbonyl compounds as ketones; either combine these two groups or change the "Total carbonyl compounds" in "Total ketones".

- Please place the "nd: not detected" in table footnote, not as part of the table.

Otherwise, the results were explained and accompanied with references of previous studies. Conclusion summarises all and opens possibilities for further research.

Good luck!

Author Response

Dear authors,

your manuscript deals with an interesting subject, and you did a big research. Therefore, it should be presented in its best quality. I have few recommendation:

- In the abstract, line 13/14 - after Saccharomyces cerevisiae (SC), I recommend adding: "as reference strain".

Done as requested.

When reading only abstract, a reader could ask how is this non-Saccharomyces yeast, while the title indicates that the study is only about them.

We believe that, thanks to the previous comment of reviewer 2, this point is clearer now.

- I recommend adding some information about thermally inactivated yeasts; why? how?, so readers could better understand the purpose of the study.

As reported in lines 51-53, non-Saccharomyces thermally inactivated yeasts (TIYs) release soluble compounds, like polysaccharides, lipids, thiols and reduced glutathione (GSH) and show different oxygen consumption rate and antioxidant activity in a wine-like solution. For this reason, we decided to focus on these particular yeast derivatives. How these derivatives have been obtained is reported in lines 70-72.

- Phenolic compounds determination by HPLC: Is there a reason why no individual phenolic compounds were identified with HPLC, only the total sum?

We inserted a supplementary table (Table 1s) reporting all the compounds detected in the white wines analyzed. We identified most of the compounds  by both injection of chemical standards and UV-visible spectra obtained with DAD but, as you will see, some of them are unknown. In Table 3  we reported the total phenols obtained as the peaks areas sum at 280 nm. The relevant  comments are reported in the manuscript.

- Volatile compounds determination by GC-MS: Could you please explain what was the peak identification method (retention time, retention index, or mass spectra base)?

Peaks identification was based on retention time and electron impact mass spectra.

- How did you calculate the concentrations of volatile compounds? Was the response ratio for each compound calculated in comparison to the internal standard, or the calculation was tentatively using only peak area and concentration of internal standard? This could also be added in the text if possible.

For each compound pure standard have been used and a calibration curve was built based on the response ratio in comparison to the internal standard. For quantification the quantifier ion in SIR mode was selected.

- Table 4. Volatile compounds: Why only total aldehydes are presented, and no individual compounds? How was this obtained?

Benzaldehyde was the only aldehyde detected. We change Table 4 accordingly.

- Aldehydes are also classified as carbonyl compounds as ketones; either combine these two groups or change the "Total carbonyl compounds" in "Total ketones".

We thank reviewer 2 for this observation. Based on that table 4 was reorganized and Figure 3 was also updated accordingly.

- Please place the "nd: not detected" in table footnote, not as part of the table.

Since acetoin was deleted from table 4, because no detectable, the legend of table 4 was changed accordingly.

Otherwise, the results were explained and accompanied with references of previous studies. Conclusion summarises all and opens possibilities for further research.

Good luck!

Reviewer 3 Report

Comments and Suggestions for Authors

The manuscript investigated the impact of TIYs derived from Saccharomyces cerevisiae, Saccharomycodes ludwigii, Metschnikowia pulcherrima and Torulaspora delbrueckii on total polysaccharides, heat test, color, total phenol and volatile compounds of white wine Trebbiano Toscano. The experimental design is good, and the results are very interesting. However, the wine information should be shown clearly.

Line 71 Are wines just after fermentation, or after aging of a long time ?

Line 74-75 How much wine were treated?

Line 184-185 All the TIYs except MP-TIY

Author Response

The manuscript investigated the impact of TIYs derived from Saccharomyces cerevisiae, Saccharomycodes ludwigii, Metschnikowia pulcherrima and Torulaspora delbrueckii on total polysaccharides, heat test, color, total phenol and volatile compounds of white wine Trebbiano Toscano. The experimental design is good, and the results are very interesting. However, the wine information should be shown clearly.

Line 71 Are wines just after fermentation, or after aging of a long time ?

The wine utilized was not subjected to stabilization treatments or aging. This information was added to the text.

Line 74-75 How much wine were treated?

All the trials were carried out on 200 mL samples. This information was added to the text.

Line 184-185 All the TIYs except MP-TIY

We thank reviewer 3. We change the sentence accordingly.

Round 2

Reviewer 2 Report

Comments and Suggestions for Authors

Dear authors,

Thank you for all your answers and for accepting my main suggestions.

I have no further comments and therefore it is my opinion that this paper could be submited in this journal.

Best regards.